# Microfluidic Distillation System for Separation of Propionic Acid in Foods

**DOI:** 10.3390/mi14061133

**Published:** 2023-05-28

**Authors:** Song-Yu Lu, Chan-Chiung Liu, Kuan-Hsun Huang, Cheng-Xue Yu, Lung-Ming Fu

**Affiliations:** 1Department of Engineering Science, National Cheng Kung University, Tainan 70101, Taiwan; n98091055@gs.ncku.edu.tw (S.-Y.L.); n96104496@gs.ncku.edu.tw (K.-H.H.); hank15861586@gmail.com (C.-X.Y.); 2Department of Food Science, National Pingtung University of Science and Technology, Pingtung 91201, Taiwan; ccliu@mail.npust.edu.tw; 3Graduate Institute of Materials Engineering, National Pingtung University of Science and Technology, Pingtung 91201, Taiwan

**Keywords:** Propionic acid, micro-distillation, microfluidics, food

## Abstract

A microfluidic distillation system is proposed to facilitate the separation and subsequent determination of propionic acid (PA) in foods. The system comprises two main components: (1) a polymethyl methacrylate (PMMA) micro-distillation chip incorporating a micro-evaporator chamber, a sample reservoir, and a serpentine micro-condensation channel; and (2) and a DC-powered distillation module with built-in heating and cooling functions. In the distillation process, homogenized PA sample and de-ionized water are injected into the sample reservoir and micro-evaporator chamber, respectively, and the chip is then mounted on a side of the distillation module. The de-ionized water is heated by the distillation module, and the steam flows from the evaporation chamber to the sample reservoir, where it prompts the formation of PA vapor. The vapor flows through the serpentine microchannel and is condensed under the cooling effects of the distillation module to produce a PA extract solution. A small quantity of the extract is transferred to a macroscale HPLC and photodiode array (PDA) detector system, where the PA concentration is determined using a chromatographic method. The experimental results show that the microfluidic distillation system achieves a distillation (separation) efficiency of around 97% after 15 min. Moreover, in tests performed using 10 commercial baked food samples, the system achieves a limit of detection of 50 mg/L and a limit of quantitation of 96 mg/L, respectively. The practical feasibility of the proposed system is thus confirmed.

## 1. Introduction

Propionic acid (PA) (CH_3_CH_2_COOH) is a three-carbon short-chain fatty acid formed naturally in the human body through the fermentation of dietary fiber and indigestible carbohydrates by symbiotic bacteria in the colon. It is also widely used in the pesticide, food, plastics, and beverage industries. One of the most common uses of PA is as a preservative in extending the shelf life of baked foods such as bread and cookies.

PA is naturally present in the human body and plays an important role in preventing obesity and improving the health condition of diabetes type 2 patients [1,2]. However, an excessive intake of PA is associated with a range of adverse health effects, including cognitive decline [2], gingival inflammation [3], neurotoxicity [4], and the aggravation of autism spectrum disorders (ASD) [5]. Consequently, the concentration of PA additives in foods and beverages must be carefully controlled. For example, the Taiwan Food and Drug Administration (TFDA) stipulates that the concentration of PA in bread and cakes should not exceed 2.5 g/kg (2500 ppm).

Many methods are available for separating and quantifying the PA content in food, including high-performance liquid chromatography-UV detection (HPLC-UV detection) [6], gas chromatography-mass spectrometry (GC/MS) [7], and gas chromatography-flame ionization detection (GC-FID) [8]. These methods all require sample pretreatment prior to the separation and detection process. Owing to commonly high fat content and volatile components, the PA in baked foods must be isolated in some way such that its content can be properly determined. The HPLC method is most commonly performed using steam distillation (see Official Method No. 1001900044TFDA of the Taiwan Food and Drug Administration (TFDA), for example). However, the distillation process is lengthy (typically around 4~6 h) and requires professional expertise and the use of bulky and specialized equipment. Thus, the distillation procedure is largely confined to modern and well-equipped laboratories. Consequently, the development of alternative distillation methods which allow the PA pretreatment process to be performed in a cheaper, faster, and more straightforward manner is of great interest.

Microfluidics devices have many advantages, such as a lower sample and reagent consumption, cheaper cost, simpler operation, faster detection time, and improved sensitivity [9,10,11,12,13,14,15,16,17,18]. As a result, they have found widespread uses in many fields nowadays, including biomedical analysis, food safety screening, drug development, and environmental monitoring [19,20,21,22,23,24,25,26,27]. Many micro-distillation systems have been proposed in recent years [28,29,30,31,32,33,34,35]. Giordano et al. [29] presented a gravity-assisted micro-distillation system consisting of a polydimethylsiloxane (PDMS) microchip incorporating a heating resistor, a distillation flask, a condenser, and a distillate collector. The feasibility of the proposed system was demonstrated by performing desalination tests under harsh saline conditions. Moreover, the practicality of the system was shown by quantifying the concentration of ethanol in seven alcoholic beverages. Hsu et al. [33] proposed a micro-distillation system for quantifying the concentration of formaldehyde (CH_2_O) in food samples. The sample was vaporized by a steam flow and driven through a cooled condenser zone. The CH_2_O content of the resulting distillate was then determined using an AHMT (4-Amino-3-hydrazino-5-mercapto-1,2,4-triazole, C_2_H_6_N_6_S)-based colorimetric spectrometry technique. The results showed that the system achieved an efficiency of around 98%.

The performance of the micro-distillation systems described above is fundamentally dependent on the heat transfer efficiency and temperature distribution within the microfluidic device. Accordingly, many researchers have employed numerical simulation methods to optimize the performance of micro-distillation systems [36,37,38,39,40,41,42]. For example, Stanisch et al. [38] conducted numerical simulations to investigate the effects of the primary processing parameters (e.g., the reflux ratio, evaporation rate, and choice of feed stage) on the performance of a micro-distillation system intended for the separation of ethanol/water feed streams. Overall, the results presented in [37,38] confirmed that numerical simulations provide a versatile and effective approach for the design, characterization, and optimization of microfluidic distillation systems.

The present study proposes a microfluidic distillation system consisting of a PMMA microchip and a self-built distillation module for separating the PA content in foods and beverages. The microchip incorporates a micro-evaporator filled with deionized (DI) water, a sample chamber, a serpentine microchannel condenser, and a distillate collection zone. In the distillation process, the microchip is mounted on the side of the distillation module and the evaporator chamber is heated in order to produce steam. The steam flows through a connecting microchannel to the sample chamber, where it vaporizes the homogenized sample to produce PA vapor. The vapor then flows through the serpentine channel, where it is cooled and condensed to produce PA extract. A small quantity of the distilled extract is then transferred to a HPLC and photodiode array (PDA) detector system to determine the corresponding PA concentration.

## 2. Materials and Methods

### 2.1. Micro-Distillation Chip Fabrication

Figure 1a presents a schematic illustration of the proposed micro-distillation chip consisting of a PMMA cover layer (thickness 1.5 mm), a PMMA chip body layer (thickness 6 mm), and an aluminum foil adhesive layer (thickness 0.3 mm). As shown in Figure 1b,c, the body layer of the microchip incorporates an evaporation chamber, a sample reservoir, a serpentine condensation microchannel, a distillate collection zone, and a release valve. The release valve is designed to prevent excessive pressure in the micro-condensation channel during distillation, to balance the pressure between the micro-condensation channel and the atmosphere, and to prevent the extract solution stock from splashing out. The PMMA layers were designed using commercial AutoCAD software (2011) and fabricated by a CO_2_ laser ablation system [43]. The cover layer and body layer were joined using a conventional hot-press bonding technique, and the aluminum foil was adhered to the bottom of the body layer to seal the device and improve the heat transfer efficiency within the chip. The finished chip had overall dimensions of 210 mm × 76 mm × 7.8 mm.

Compared with the devices proposed in previous studies by the present group [33,34], the proposed microfluidic distillation system has the advantage that the steam required for distillation purposes is generated by an external heating source mounted in the distillation module. Similarly, the cooling effect required to condense the PA vapor in the serpentine coil is also produced by an external system installed in the distillation module. Thus, the size, cost, and complexity of the micro-distillation chip are all reduced. Furthermore, the water required to vaporize the homogenized sample is stored in the chip itself, and hence the need for an external water tank is removed. Finally, the simple design of the distillation chip, together with its low cost (<US$3), renders it suitable for single-use application, thereby eliminating the risk of cross-contamination from samples.

In general, different food additives have different characteristics (e.g., different boiling points, densities, chemical properties, and so on) and thus appropriate micro-distillation chip designs are required to maximize the separation efficiency depending on a specific analyte. The simple design of the microchip proposed in the present study lends itself to the use of numerical simulation methods, optimizing not only the design of the micro-distillation chip but also the operating conditions.

### 2.2. Distillation Module

Figure 2a shows the self-built distillation module developed in the present study. As shown, the main components include a power supply system, a heater module, a cooler module, and two temperature control panels. The heater module comprised a 15 W pliability heater (TSC0100010gR70.5, King Lung Chin Co., Ltd., Taichung, Taiwan) with a maximum temperature capability of 180 °C mounted on a solid copper block with dimensions of 52 mm × 30 mm × 10 mm. The cooling block consisted of a commercial cooling module (72041/071/150B, Ferrotec Taiwan Co., Ltd., Hsinchu, Taiwan) with a power of 10 W, a minimum temperature capability of 4 °C, and a size of 110 mm × 40 mm × 10 mm. The module casing was made of ABS using a 3D printer (Kingssel K3040, Mastech Machine Co., Ltd., New Taipei City, Taiwan) and measured 200 mm × 100 mm × 65 mm.

As shown in Figure 2b, in the distillation process, the micro-distillation chip was clipped to the side of the micro-distillation module and positioned such that the evaporator chamber and micro-condensation channel were aligned with the heater and cooler modules, respectively. On completion of the distillation process, the device was removed from the module and a small quantity of distillate was retrieved from the collection zone and transferred to a cuvette for HPLC-PDA determination of the PA concentration.

### 2.3. Experimental Details

The reagents employed in the present study included phosphoric acid (H_3_PO_4_, 85~87%, J. T. Baker, Phillipsburg, NJ, USA), PA (CH_3_CH_2_COOH, Nippon Reagent Industry Co., Ltd., Osaka, Japan, boiling point: 141 °C), and ammonium dihydrogen phosphate ((NH_4_)H_2_PO_4_, Showa Kako Corp., Osaka, Japan). All the chemicals were reagent grades with a resistance of 18.2 MΩ in DI water. A 1 M phosphoric acid solution was prepared by diluting 67.4 mL phosphoric acid in 1000 mL DI water. 1 g of PA was dissolved in 100 mL of DI water and then diluted with 1 M phosphoric acid solution as required to produce control samples with concentrations of 50~3000 mg/L. 1.5 g of diammonium hydrogen phosphate was dissolved in 1000 mL of DI water and adjusted to pH 3 through the addition of phosphoric acid to serve as the mobile phase for the HPLC determination process.

To determine the PA concentration of the real food samples, 5 g of each food was homogenized by a commercial machine, and 0.1 g of the homogenized sample was dissolved in 1 mL DI water for 15 min distillation in the micro-distillation chip. Following the distillation process, the PA content of the sample was determined via HPLC-PDA system in accordance with the official method published by the Taiwan Food and Drug Administration (TFDA, No. 1001900044). For comparison, the PA content of the food samples was also evaluated using a traditional benchtop steam distillation apparatus followed by HPLC-PDA separation and detection in accordance with the TFDA No. 1001900044 method. In the distillation process, the temperature and cooling modules of the distillation unit were set to 150 °C and 20 °C, respectively, and 4.5 mL of DI water was injected into the micro-evaporation chamber of the chip. 1 mL of homogenized sample (containing 0.1 g of the original sample) was placed in the sample reservoir. The injection inlets of the evaporation chamber and sample chamber were both sealed with heat-resistant tape. The microchip was then clipped to the side of the distillation module (as shown in Figure 2b). During the distillation process, the steam produced in the evaporation chamber flowed into the sample chamber, prompting the generation of PA vapor. The vapor flowed through the cooled serpentine channel, where it condensed and then entered the distillate collection zone due to gravity and steam driving force. The distillation process was stopped after 15 min (as discussed later in Section 3). The distillate was retrieved from the collection zone, and its pH value was adjusted to about 3.0 through the addition of 1 M phosphoric acid solution. 25 µL of the test solution (according to TFDA, No. 1001900044) was taken for HPLC-PDA analysis. The HPLC procedure was conducted on a Shimazdzu LC-20AT system equipped with a 5-μm reversed-phase chromatography column (Agilent ZORBAX Eclipse Plus C18, 0.46 cm × 25 cm). The separation process was performed using 0.15% disodium hydrogen phosphate as the mobile phase with a flowrate of 1.2 mL/min. PDA detection was then performed using ultraviolet light with a wavelength of 214 nm.

## 3. Results

In general, numerical simulations provide an efficient means of optimizing the design of micro-distillation chips and exploring the corresponding flow field, steam temperature, and distillation efficiency [44,45]. In the present study, the flow field and steam temperature distribution within the micro-distillation chip were examined by ANSYS FLUENT simulations. (Note that full details of the numerical method and solution procedure are described elsewhere [33,34].) As shown in Figure 3a, a vortex structure was formed as the vapor stream entered the sample reservoir after being accelerated through the connecting microchannel. The vortex structure perturbed the sample within the chamber, thereby improving the vaporization efficiency. Figure 3b shows the simulated temperature distribution within the microchip. In general, the results confirm that a temperature setting of 150 °C for the micro-evaporator chamber is sufficient to prompt the vaporization of the PA, while a cooling temperature of 20 °C is sufficient to condense the vapor and produce PA distillate in the collection zone.

Overall, the simulation results substantiate the ability of the micro-distillation chip to accomplish the distillation and condensation operations required, to separate the PA content of the homogenized sample prior to HPLC-PDA determination. The average temperatures of the sample reservoir and distillate outlet of the micro-distillation chip were measured experimentally using thermocouples and were found to be 101.8 °C and 20.5 °C, respectively. The simulated temperature values (i.e., 100.2 °C and 20.1 °C, respectively) deviated by no more than 2.5% from the experimental measurements. Thus, the basic validity of the numerical model was confirmed.

For calibration purposes, six control solutions with known PA concentrations in the range of 50~3000 mg/L were prepared. For each sample, the distillation (separation) efficiency was evaluated by Equation (1).
(1)$$Distillation\;\left(separation\right)\;efficiency=\frac{Distilled\;of\;PA\left(mg/L\right)}{References\;of\;PA\left(mg/L\right)\;}$$

Figure 4 shows the variation in the distillation efficiency over time for the control sample with a PA concentration of 1000 mg/L. As shown, the efficiency increases initially with an increasing distillation time, which could be attributed to increased amount of steam produced in the evaporation chamber due to increasing distillation time. Thus, a greater amount of acid vapor is generated in the sample chamber and flows through the condensation channel. However, as the heating time is further increased, the DI water in the evaporation chamber is gradually consumed. Consequently, the quantity of PA vapor reduces, and the distillation efficiency saturates at an approximately constant value. The maximum distillation efficiency is around 97% and is obtained after 15 min. Consequently, the distillation time was set as 15 min in all of the remaining distillation experiments.

Figure 5 shows the experimental results for the distillation efficiencies varied with the condensation channel length. As the channel length first increases, the distillation efficiency also increases since the time for which the PA vapor is exposed to the low temperature condition (20 °C) increases. However, as the channel length increases beyond 50 cm, the driving force provided by the steam is insufficient to push the distillate through the channel and into the collection zone, and therefore the distillation efficiency drops. Accordingly, the optimal condensation channel length was determined to be 50 cm.

Figure 6 shows the variation in the distillation efficiency with the volume of DI water injected into the micro-evaporator chamber of the chip. Note that the results correspond to the 1000 mg/L control sample with a volume of 1 mL. As the amount of DI water increases, the volume of steam vapor generated over the distillation process also increases, hence a greater amount of distillate is obtained in the collection zone. However, for 5 mL of water, the entire sample is distilled within 15 min and accompanied by saturation of distillation efficiency. Accordingly, the injection volume of DI water was set as 4.5 mL and the sample volume as 1 mL in all of the remaining experiments.

The feasibility of the proposed microfluidic distillation system was investigated by measuring the PA concentrations of the six control samples with known concentrations of 50 mg/L, 500 mg/L, 1000 mg/L, 1500 mg/L, 2500 mg/L and 3000 mg/L, respectively. For comparison, the concentrations were also measured using the official HPLC-PDA detection method with a benchtop steam distillation apparatus.

Figure 7 compares the measurement results obtained by the two methods. The high correlation coefficient (R^2^ = 0.9971) indicates a good agreement between the two sets of results. Moreover, six different PA concentrations (50 mg/L, 500 mg/L, 1000 mg/L, 1500 mg/L, 2500 mg/L and 3000 mg/L) were added to PA-free breads. The analytical accuracy of the proposed microfluidic distillation system and HPLC-PDA detector system is 96.7 ± 1.8%. (Note that the accuracy was evaluated using Equation (1) below.) Thus, the basic feasibility of the proposed system was validated.

The practical applicability of the proposed system was verified by detecting the PA concentrations of 10 real-world baked food samples acquired from convenience stores in Taiwan (see Table 1). For each sample, the pretreatment process was performed using a microfluidic distillation system (as described in Section 2.3), and the PA content was then evaluated using the HPLC-PDA method listed in Official Method No. 1001900044 of the Taiwan Food and Drug Administration (TFDA). For comparison purposes, the PA content of each sample was also evaluated by the Center for Agriculture and Aquaculture Product Inspection and Certification (CAAPIC) at National Pingtung University of Science and Technology (NPUST) in Taiwan, using the benchtop steam distillation, separation, and detection procedures with the same official method. In the case of the micro-distillation process, the reliability of the measurement results was ensured by testing each food sample five times using a newly homogenized sample on each occasion.

As shown in Table 1, no PA was detected in Samples #4, #8, or #9 using the micro-distillation chip. Thus, it was inferred that these samples either contained no PA, or had a PA concentration lower than the limit of detection (LOD) of the proposed device. For the official method conducted by CAAPIC, no PA was detected in Samples #4, #8, or #9 or in Sample #2.

Taking the detection results obtained using the exact official HPLC method as a benchmark, the detection accuracy of the proposed micro-distillation system was quantified by Equation (2).
(2)$$\mathrm{Accuracy}=\left(1-\frac{CAAPIC\;Method-Microdistillation\;method)}{CAAPIC\;Method}\right)\times 100\%$$

As shown in Table 1, the detection accuracy varies from 96.8% (Sample 10) to 99.2% (Sample 3). In other words, the accuracy deviates from that of the official method by no more than 3.2%. Moreover, the proposed method has a LOD of 50 mg/L and a LOQ of 96 mg/L. Finally, the proposed distillation method requires just 0.1 g of sample for determination purposes, whereas the exact official method requires more than 100 g. Thus, the proposed system has significant benefits over the traditional method for the real-world determination of the PA concentration in baked food products.

Table 2 presents a qualitative comparison of the microfluidic distillation system and detection method proposed in the present study with other PA detection methods reported in the literature. The micro-distillation method developed in this study not only achieves an outstanding recovery, but also could shorten the sample pretreatment time significantly. Moreover, the developed method requires only a very small amount of test sample and analytical reagents. All the advantages combined would make this developed microdistillation chip a perfect tool in a largescale market sampling survey of PA in foods.

## 4. Conclusions

This study has presented a microfluidic distillation system to facilitate the determination of the PA concentration in baked foods. The proposed system consists of a PMMA-based micro-distillation chip and a self-built distillation module with heating and cooling components. By retrieving the distillate from sample, the PA concentration is determined using a conventional HPLC-PDA system. The proposed microfluidic distillation system provides several important advantages over a traditional benchtop apparatus, including a higher throughput, a reduced sample and reagent consumption, a lower power consumption, minimal risk of cross-contamination, greater portability, and a lower fabrication cost.

The experimental results have shown that the microfluidic distillation system achieves a distillation efficiency of 97% in 15 min. Moreover, the detection results obtained for control samples with known PA concentrations in the range of 50~3000 mg/L have been shown to be in excellent agreement (R^2^ = 0.9971) with those obtained using an official HPLC-PDA detection method with a traditional benchtop steam distillation process. Finally, the detection results obtained for 10 real-world baked food products have shown that the proposed system has an LOD of 50 mg/L and an LOQ of 96 mg/L. The system thus outperforms the official distillation method employed in the present study (LOQ = 500 mg/L) and provides a rapid and feasible approach for practical PA determination in foods.

## Figures and Tables

**Figure 1 micromachines-14-01133-f001:**
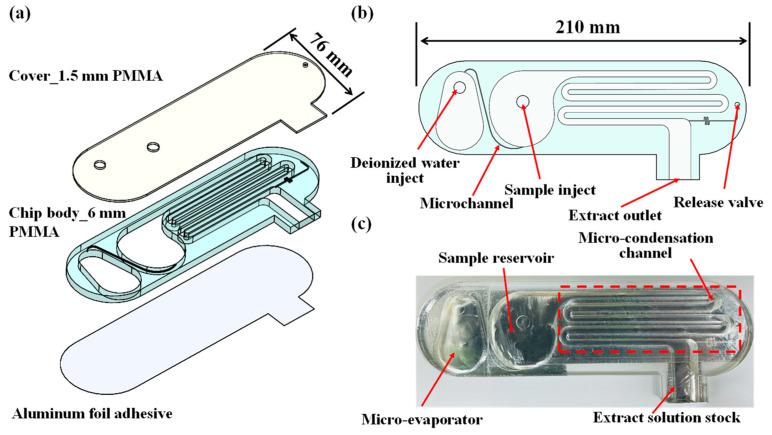
Configuration and dimensions of micro-distillation chip: (**a**) dimensions and layout of individual layers in microchip, (**b**) functional components in microchip, and (**c**) photograph of assembled chip.

**Figure 2 micromachines-14-01133-f002:**
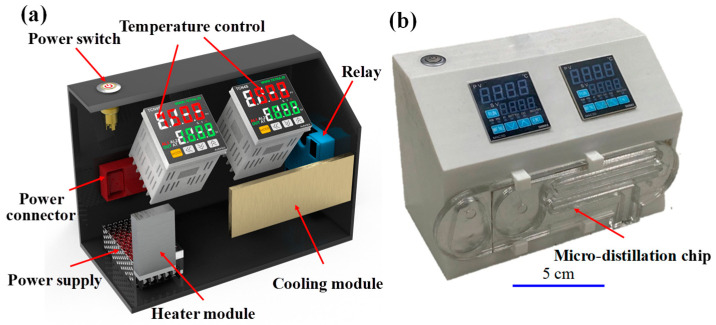
Self-built distillation module: (**a**) arrangement of main components in module, and (**b**) attachment of micro-distillation chip to side of module during distillation process.

**Figure 3 micromachines-14-01133-f003:**
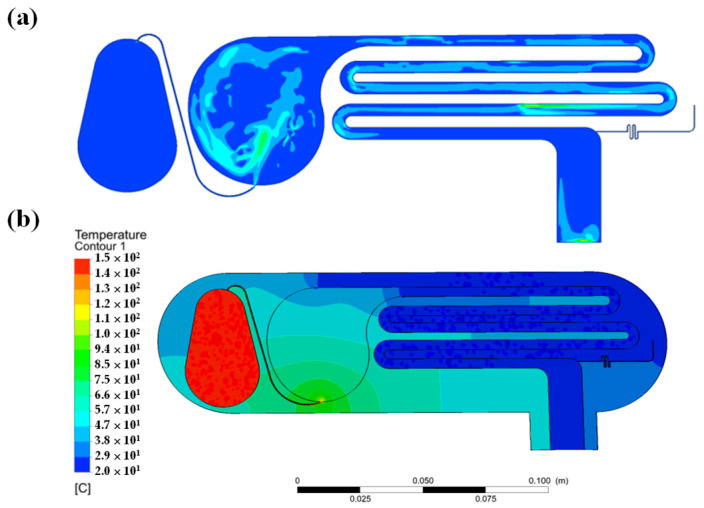
CFD simulation results for: (**a**) flow field distribution in micro-distillation chip, and (**b**) temperature distribution in micro-distillation chip.

**Figure 4 micromachines-14-01133-f004:**
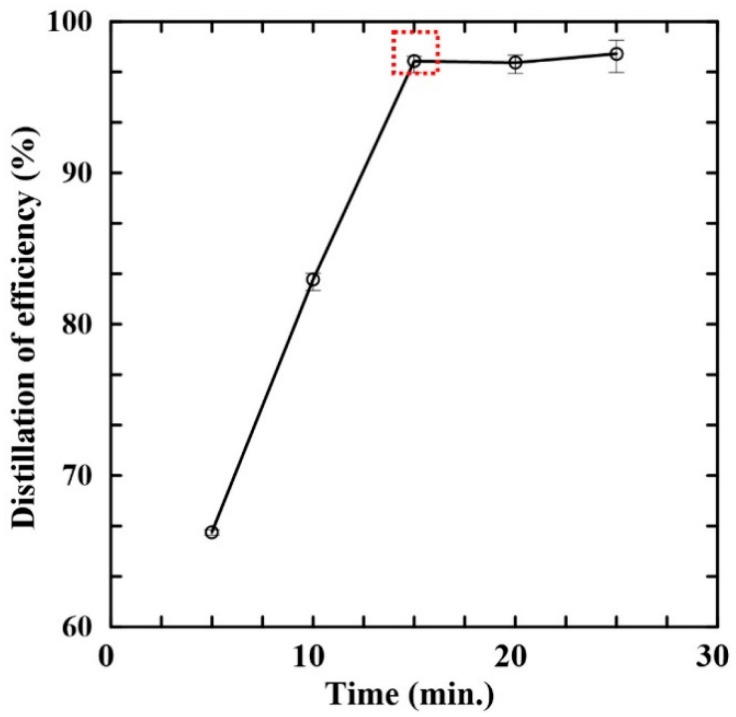
Variation in distillation efficiency over time for sample containing 1000 mg/L of PA. (Experimental operation conditions: condensation channel length: 50 cm; sample volume: 1 mL; DI water volume: 4.5 mL; distillation and condensation temperatures: 150 °C and 20 °C).

**Figure 5 micromachines-14-01133-f005:**
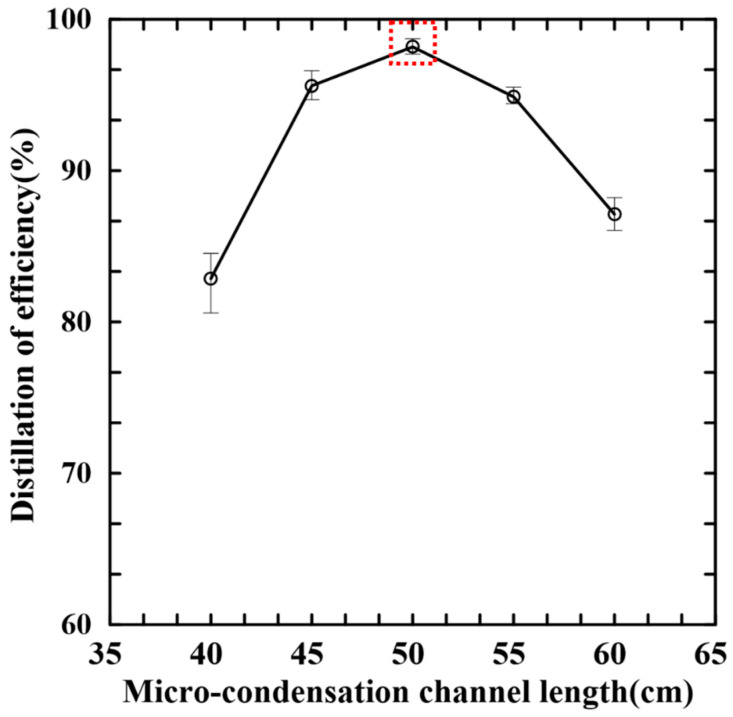
Variation in distillation efficiency with length of micro-condensation channel. (Experimental operation conditions: distillation time: 15 min; sample volume: 1 mL; DI water volume: 4.5 mL; distillation and condensation temperatures: 150 °C and 20 °C).

**Figure 6 micromachines-14-01133-f006:**
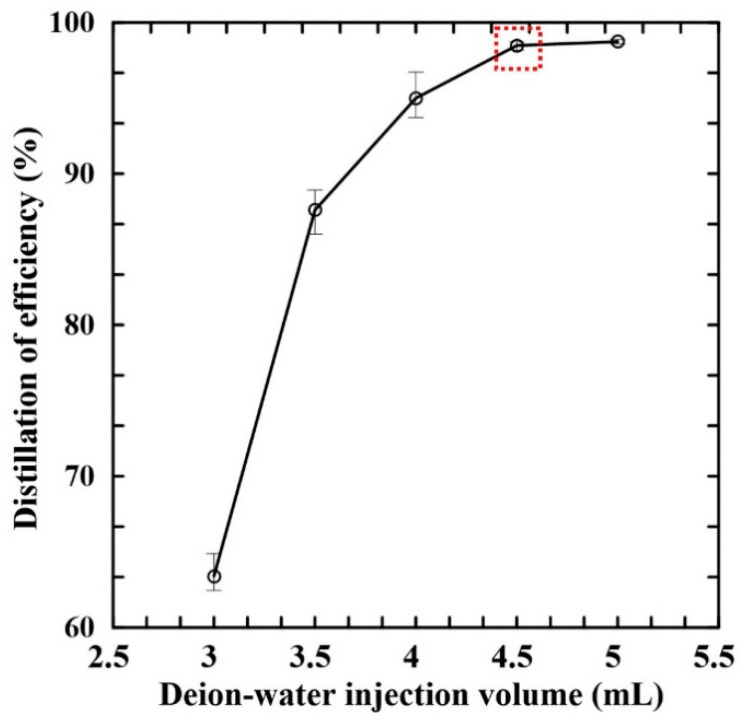
Variation in distillation efficiency with volume of DI water in evaporation chamber. (Experimental operation conditions: distillation time: 15 min; condensation channel length: 50 cm; sample volume: 1 mL; distillation and condensation temperatures: 150 °C and 20 °C).

**Figure 7 micromachines-14-01133-f007:**
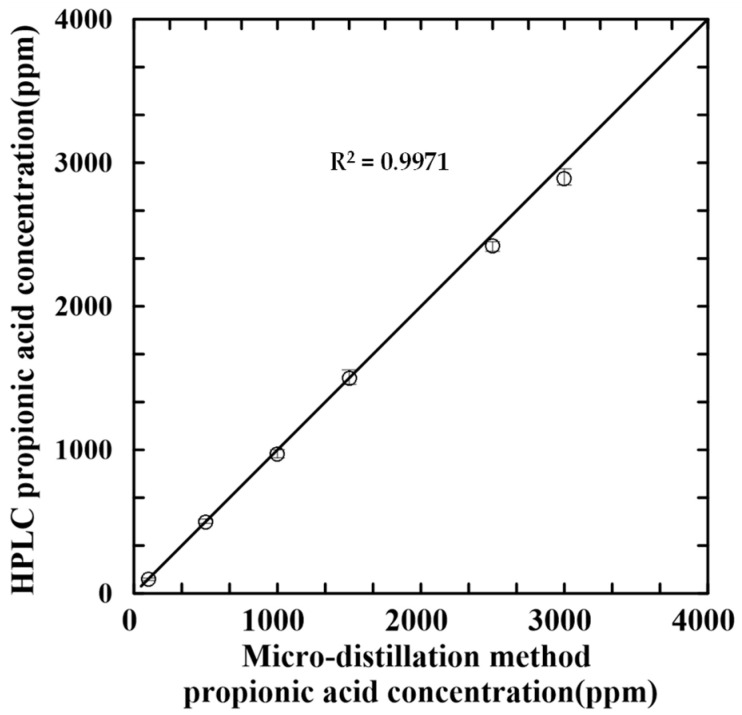
Comparison of HPLC-PDA detection results (official distillation method) and HPLC-PDA detection results (microfluidic distillation method) for control samples with known PA concentrations. (Micro-distillation method experimental operation conditions: distillation time: 15 min; condensation channel length: 50 cm; sample volume: 1 mL; DI water volume: 4.5 mL; distillation and condensation temperatures: 150 °C and 20 °C).

**Table 1 micromachines-14-01133-t001:** Comparison of detection results obtained by two methods for 10 commercial baked food products. (The detection values reported for the microfluidic distillation technique represent the average values obtained over five independent measurements).

Sample No.	Micro-Distillation (ppm)	NPUST CAAPIC Detection (ppm)	Accuracy (%)
1 Maple Sugar Cookies	791 ± 8	800	98.9%
2 Milk cookies	92 ± 2	N. D.	—
3 Peanut cookies	1091 ± 6	1100	99.2%
4 Cookies	N. D.	N. D.	—
5 Chocolate cookies	683 ± 8	700	97.5%
6 Apple bread	505 ± 3	500	99%
7 Beard	1188 ± 5	1200	99%
8 Butter bread	N. D.	N. D.	—
9 Toast	N. D.	N. D.	—
10 whole wheat bread	581 ± 7	600	96.8%

N. D.: Non-Detectable.

**Table 2 micromachines-14-01133-t002:** Comparison of proposed microfluidic distillation–based HPLC-PDA detection method and other reported methods in the literature.

Refs.	Target	Sample Pretreatment	Separation Method	Detection Method	Limit of Detection	Time	Sample Volume
[7]	PA, BA, SA	HS-SPME	GC	MS	0.1 mg/L	65 min	20 g
[31]	PA	Microchannel distillation	Microchannel	Acid-base titration	-	1 h	-
[43]	SA, BA, SO_2_	Micro-distillation	Microchannel	Micro-spectrometer	200 mg/L	15 min	0.1 mL
[46]	PA, BA, SA	Solvent extraction	GC	MS	29.9 mg/L	49 min	100 g
[47]	PA, AA, FA, BuA	Traditional steam distillation	-	IEC	0.3 mg/L	31 min	50 mL
[48]	PA	Degrease and direct extraction	GC	FID	3 mg/L	24 min	2 g
[49]	PA	Direct extraction	GC	FID	120 mg/L	100 min	5 g
[50]	SBA	Micro-distillation	Microchannel	Colorimetric	50 mg/L	12 min	0.5 g
TFDA official method	PA	Traditional steam distillation	HPLC	PDA	-	60 min	25 g
This study	PA	Micro-distillation	HPLC	PDA	50 mg/L	15 min	1 mL

AA: acetic acid; BA: benzoic acid; BuA: butyric acid; FA: formic acid; HS-SPME: headspace solid-phase microextraction; FID: flame ionization detector; GC: gas chromatography; ICE: ion-exclusion chromatography; *MS:* mass spectrometer; SA: sorbic acid; SBA: sodium benzoate.

## Data Availability

The data presented in this study are available upon request from the corresponding author.

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
