# Peer review of "Microfluidic Distillation System for Separation of Propionic Acid in Foods"

_micromachines, 2023, doi:10.3390/mi14061133_

Round 1
Reviewer 1 Report
In this paper, the authors presented a microfluidic distillation system to facilitate the separation and subsequent determination of propionic acid (PA) in foods. The experimental results show that the microfluidic distillation system achieved a distillation efficiency of around 97% after 15 min, it outperforms the official distillation method employed in the present study (LOQ = 500 mg/L) and provides a rapid and feasible approach for practical PA determination in foods with practical feasibility. Overall, the article is comprehensive, the structure is reasonable and the experiments are complete, suggesting to accept in the present form.
Author Response
Reply: Appreciate very much for reviewer’s complimentary remarks.
Reviewer 2 Report
The manuscript is well prepared and has technical sounds. Several issues must be addressed before the manuscript can be accepted for publication.
1. The design detail of the detection system should be further described.
2. The possible pretreatment or processing procedures for various food should be discussed in the manuscript.
3. If possible, the separation efficiency should be further characterized.
The quality of English language is good.
Author Response
The manuscript is well prepared and has technical sounds. Several issues must be addressed before the manuscript can be accepted for publication.
1. The design detail of the detection system should be further described.
Reply: Thank you for your inquiry. This study consists of two main components: the microfluidic separation system and the detection system. PA was separated from baked foods using a microfluidic distillation system, and finally the PA concentration was determined using a conventional benchtop HPLC-PDA system. For the detailed introduction and design of the microfluidic separation system, please refer to Section 2.1~2.2. Also, details of the conventional benchtop HPLC-PDA detection system setup could be referred to Section 2.3. (P.3~P.5)
2. The possible pretreatment or processing procedures for various food should be discussed in the manuscript.
Reply: Thank you for your question. This study focuses on the microfluidic separation of PA in baked foods. The pretreatment process and processing procedures of baked foods in this study followed the guidelines proclaimed by TFDA No. 1001900044 (Taiwan Food and Drug Administration). More details on the pretreatment procedure and processing procedures could be found in section 2.3 of this manuscript. (P.4~P.5)
3. If possible, the separation efficiency should be further characterized.
Reply: Thanks to the reviewer. In fact, the separation efficiency of the current microfluidic distillation system is designated as distillation efficiency. We have added this information in the Abstract and Eq. (1). (P.1 and P.6)
Reviewer 3 Report
The authors reported a microfluidic distillation system consisting of a PMMA microchip and a self-built distillation module for separating the PA content in foods. Compared with traditional method, this microfluidic system achieves a distillation efficiency of 97% in 15 min. I recommend the minor revision of this paper, to adjust the structure of the article and solve the following issues:
1. Whether the distillated sample would be stuck due to the air pressure in condensation channel? The authors should introduce more clearly about the use of the release valve. How do they set up the open/close time?
2. The authors only considered the length of the condensation channel, does the shape or the structure affect the distillation efficiency?
3. How do the authors calculate LOD and LOQ of the proposed method?
4. More HPLC-PDA results should be introduced and analyzed. I am wondering whether the distillated sample is pure.
5. Do the authors consider adding the analyzing part into the microfluid device? As we know HPLC is still a time-consuming analytical chemistry method. Directly analyzing PA in the microfluidic chip may provide a more convenient solution.
Minor editing of English language required.
1. Line 335, “R2” should be “R2”.
Author Response
The authors reported a microfluidic distillation system consisting of a PMMA microchip and a self-built distillation module for separating the PA content in foods. Compared with traditional method, this microfluidic system achieves a distillation efficiency of 97% in 15 min. I recommend the minor revision of this paper, to adjust the structure of the article and solve the following issues:
- Whether the distillated sample would be stuck due to the air pressure in condensation channel? The authors should introduce more clearly about the use of the release valve. How do they set up the open/close time?
Reply: Thank you for your question.
(1) In fact, to determine the PA concentration of the real food samples, 5 g of each food was homogenized by a commercial machine, and 0.1 g of the homogenized sample was dissolved (mixed) in 1 mL DI water for distillation in the micro-distillation chip. The solid samples (baked foods) were completely homogenized and blended with DI water, thus forming a very dilute slurry solution. According to the results of many experiments, the sample slurry during distillation had not stuck in the condensation channel.
(2) In order to prevent excessive pressure in the micro-condensation channel during distillation, a release valve is designed to balance the pressure between the micro-condensation channel and the atmosphere to prevent the extract solution stock from splashing out. The release valve does not need to be set on/off, but is always on.
We also added a paragraph to Section 2-1 to explain the function of the release valve. (P.3)
“The release valve is designed to prevent excessive pressure in the micro-condensation channel during distillation, to balance the pressure between the micro-condensation channel and the atmosphere to prevent the extract solution stock from splashing out.”
- The authors only considered the length of the condensation channel, does the shape or the structure affect the distillation efficiency?
Reply: Thank you for your question. In fact, we have numerically simulated micro-condensation channel shapes including square, triangular and circular channels. As a result, the distillation efficiency is only related to the length of the micro-condensation channel under the same flow rate, while the shape and structure of the micro-condensation channel do not affect the distillation efficiency.
- How do the authors calculate LOD and LOQ of the proposed method?
Reply: LOD (limit of detection) was calculated as LOD = 3.3 σ/S, and LOQ (limit of quantification ) = 10 σ/S. σ is the standard deviation of the response (integrated area of HPLC-PDA, arbitrary unit) obtained for lowest known concentration PA, and S is the slope of the calibration curve established by 6 control solutions with known PA concentrations in the range of 100 ~ 3000 mg/L (unit in reverse of mg/L).
- More HPLC-PDA results should be introduced and analyzed. I am wondering whether the distillated sample is pure.
Reply: Thank you for your question. The separation and detection of PA in food can be carried out by direct extraction and steam distillation according to the official method of Taiwan Food and Drug Administration (TFDA, No. 1001900044). However, baked foods commonly have a high fat content and many volatile components, which cannot be extracted directly. They can only be separated by steam distillation to effectively remove the influence of other interfering substances. The final detection was performed by HPLC-PDA method (see, TFDA, No. 1001900044). In this study, we separated PA from baked foods using a microfluidic distillation system to obtain a PA solution free of other interfering substances.
- Do the authors consider adding the analyzing part into the microfluid device? As we know HPLC is still a time-consuming analytical chemistry method. Directly analyzing PA in the microfluidic chip may provide a more convenient solution.
Reply: In fact, our research group is already developing a paper-based detection platform and a microfluidic chip detection platform for PA concentration detection. It can be combined with a microfluidic distillation system to accomplish the rapid separation and detection of PA concentration.
- Line 335, “R2” should be “R2”.
Reply: Thank you for your reminder. The authors revise the word.
Round 2
Reviewer 2 Report
The authors have addressed all the questions I raised. The manuscript can be accepted as it.